# Antithrombin and Its Role in Host Defense and Inflammation

**DOI:** 10.3390/ijms22084283

**Published:** 2021-04-20

**Authors:** Christine Schlömmer, Anna Brandtner, Mirjam Bachler

**Affiliations:** 1Department of Cardiac Thoracic Vascular Anaesthesia and Intensive Care Medicine, Medical University of Vienna, 1090 Vienna, Austria; christine.schloemmer@meduniwien.ac.at; 2Division of General and Surgical Critical Care Medicine, Department of Anesthesiology and Critical Care Medicine, Medical University of Innsbruck, 6020 Innsbruck, Austria; mirjam.bachler@i-med.ac.at

**Keywords:** antithrombin, host response, antimicrobial peptides, anticoagulant

## Abstract

Antithrombin (AT) is a natural anticoagulant that interacts with activated proteases of the coagulation system and with heparan sulfate proteoglycans (HSPG) on the surface of cells. The protein, which is synthesized in the liver, is also essential to confer the effects of therapeutic heparin. However, AT levels drop in systemic inflammatory diseases. The reason for this decline is consumption by the coagulation system but also by immunological processes. Aside from the primarily known anticoagulant effects, AT elicits distinct anti-inflammatory signaling responses. It binds to structures of the glycocalyx (syndecan-4) and further modulates the inflammatory response of endothelial cells and leukocytes by interacting with surface receptors. Additionally, AT exerts direct antimicrobial effects: depending on AT glycosylation it can bind to and perforate bacterial cell walls. Peptide fragments derived from proteolytic degradation of AT exert antibacterial properties. Despite these promising characteristics, therapeutic supplementation in inflammatory conditions has not proven to be effective in randomized control trials. Nevertheless, new insights provided by subgroup analyses and retrospective trials suggest that a recommendation be made to identify the patient population that would benefit most from AT substitution. Recent experiment findings place the role of various AT isoforms in the spotlight. This review provides an overview of new insights into a supposedly well-known molecule.

## 1. Overview

Antithrombin (AT) is synthesized in the liver and, along with protein C and protein S, is one of three major endogenous anticoagulants. Its functional characteristics arise from two distinct binding sites: one interacts with the active sites of coagulation factor proteases, the second one binds to therapeutic heparin and glycosaminoglycans (GAGs) containing a 3-OS-modification on cellular surfaces [1]. Although AT alone exerts inhibitory activity, its binding to mast cell-derived heparin or endothelial GAGs, such as heparan sulfate [2] or heparinase, initiates a conformational change leading to an increase in AT activity by several orders [3,4]. AT binds to active coagulation factors of both the intrinsic and the extrinsic coagulation system, such as FIXa [5]; contact pathway factors [6], including kallikrein [7], FXIa, FXIIa, FVIIa [8]; and with highest affinity to FXa and thrombin (FIIa). Its binding results in the inactivation of the above-mentioned factors and, therefore, acts as an anticoagulant factor within the coagulation cascade (reviewed in [9]). The importance of the role of AT in the coagulation cascade is exposed in situations with antithrombin deficiency or where its function is impaired in carriers of AT mutations. For instance, the thrombin inhibition capacity by AT is significantly impaired in plasma of patients carrying mutations in the SERPINC1 gene [10]. The AT Budapest 3 (ATBp3) mutation leads to changes in the type II heparin-binding site (IIHBS) and, subsequently, to a functional AT deficiency due to decreased heparin–AT interactions. This was shown to be associated with an increased risk for thrombosis [11,12].

Since the primary function of AT is its essential role as anticoagulant, the dynamics in severe inflammatory syndromes, such as sepsis, are well researched. The lower the levels of AT, the worse the projected outcome for the septic patient [13,14].

COVID-19 is known to alter the coagulation and, in severe cases, leads to a hypercoagulatory state, which is causally involved in non-favorable patient outcomes [15,16,17]. Microthromboses of the pulmonary vasculature were described early after COVID-19 first peaked in Europe in spring 2020. It partially caused the pronounced hypoxemia during symptomatic infection [18]. Certain coagulation parameters were found to be increased, i.e., fibrinogen, D-Dimer, factor VIII, von Willebrand factor antigen, and protein C levels, whereas others were found to be decreased: AT and free protein S levels [19]. There are reports that linked mortality and decreased AT levels in COVID-19 patients [20,21]. However, decreased levels of AT are typical in severe infections, sepsis, and DIC [22]. The predictive value of AT for mortality in a population of critically ill patients was demonstrated earlier [23]. AT levels are more frequently aberrated in critically ill patients and were suggested as an indicator to advanced stages of COVID-19 [20,21].

Less prominent effects of AT concern its role in infection and immunity: AT influences the bidirectional crosstalk of inflammation and coagulation by inhibiting activated coagulation factors [24]. For example, thrombin is an enzyme that can interact with a number of receptors and proteins (recently reviewed in [25]). One of them is the family of endothelial protease-activated receptors (PARs). The activation of PARs by thrombin initiates a set of pro-inflammatory reactions but also starts regenerative processes. AT covalently binds and inactivates thrombin and prevents further downstream actions of thrombin. AT circulates in two isoforms that differ in their number of carbohydrate side-chains. Alpha AT (AT- α) has four side-chains and is the more common form, while beta AT (AT- β) has one carbohydrate group less, which makes it more affine to binding heparin [26]. In comparison to the concentration in plasma, the concentration of the β isoform of AT is higher in the extravascular compartment [27]. In addition, AT-β, together with heparin, possesses an approximately two-fold faster thrombin inhibition than does the α isoform [28]. Septic patients not only present significantly reduced AT activity but specifically a reduced concentration primarily of the β isoform [29]. As AT-β exhibits a higher binding affinity to heparin-like structures, it predominantly conveys anti-inflammatory actions [30]. The concentration of AT-β is much lower than the concentration of the α isoform (90–95%). Notably, the concentrations in commercially available AT concentrates are comparable [29].

## 2. The Role of AT in Host Response to Infection

A bacterial virulence factor is the ability of surface proteins to bind to endogenous heparin-binding molecules, such as fibronectin, to more efficiently disseminate within the host. From an evolutionary perspective, this is a most elegant strategy for the invading pathogen as there is a large array of heparin binding consensus sites throughout mammalian organisms [31]. Neisseria gonorrhoeae, for example, was shown to gain access to epithelial cells by coating with vitronectin without having a specific receptor repertoire to interact with the target cells, therefore, bypassing a certain tropism [32,33]. In fact, various Gram-positive and Gram-negative bacteria [34], *M. pneumoniae* [35], as well as. fungal [34] and viral pathogens [36], can recruit vitronectin with a heparin-binding motif. Hypothetically, AT could counteract this instrumentalization of heparin-binding proteins by bacterial structures by competitively binding to the same heparin-binding sites [31,37]. However, we speculate this would require supraphysiologic concentrations. Furthermore, it is an unexplored question whether the occupation of the target heparin-binding site by AT would attenuate the bacterial dissemination in vivo and, thus, act as a direct host response mechanism.

A direct host defense mechanism of AT is the ability to bind to LPS: recently, a compelling study by Pappareddy et al. showed direct antimicrobial effects of AT. While both AT glycosylation isoforms bind with comparable affinity to LPS on bacterial surfaces as measured by surface plasmon resonance technology, the β isoform undergoes a subsequent conformational change, which presents as AT aggregates in electron microscopy analysis [38]. This activated form of AT-β is able to perforate the bacterial cell wall and exhibits bactericidal effects against *P. aeruginosa* and *E. coli*. These bactericidal effects are based on permeabilization of cell walls, as evidenced by intracellular material found extracellularly. Both isoforms opsonized the microbes, but AT-β induced a higher phagocytic uptake by macrophages. The authors also observed that, in contrast to AT-α, the β isoform with its antimicrobial effect is depleted in patients with severe inflammation [38].

The distinct effects of AT isoforms, revealed by Pappareddy et al., are not the only antimicrobial potential of this versatile molecule. AT consumption is primarily due to formation of the protease inhibitor complex but includes a small amount of proteolytic breakdown of AT into fragments [39]. Especially during inflammatory processes, proteases, such as neutrophil elastases or bacterial proteases, are released and may also fragment AT [40].

Proteolytic AT preparations revealed that different fragments show varying degrees of bactericidal effects against different species of bacteria. One of these fragments is FFF21, whose D helix-derived peptide sequence exerts antimicrobial properties [40]. Interestingly, different fragments showed specific antimicrobial patterns: while KTS43, another AT-derived antimicrobial peptide (AMP), showed predominantly antibacterial effects, particularly against *E. coli* and *P. aeruginosa*, FFF21 and AKL22 exhibited antimicrobial effects not only against the Gram-negative bacteria *E. coli* and *P. aeruginosa* but also against the Gram-positive bacteria *S. aureus*, *B. subtilis*, and the fungi *C. albicans* and *C. parapsilosis* [40].

These findings were translated to in vivo animal models. Injecting the FFF21 peptide into mice with *P. aeruginosa* infection resulted in a significantly reduced bacterial load in various organs, such as the spleen, liver, and kidney, and also in prolonged survival [40].

A further animal experiment showed that the treatment with nebulized plasma-derived AT reduced outgrowth of *S. pneumoniae* and histopathologic damage in lungs of rats [41]. In this study, the outgrowth was reduced in the bronchoalveolar lavage fluid (BALF) of animals that were treated with nebulized AT compared to placebo.

Therefore, treatment of infections, especially with Gram-negative bacteria, with AT-derived AMPs might be a promising novel therapeutic approach since resistance to antibiotics is increasing.

The manifestation of cerebral malaria, caused by an infection with *Plasmodium falciparium,* was recently linked to a protein secreted by the parasite (histidine-rich protein II, HRPII) that competes with AT to bind to anticoagulant GAGs. Interestingly, binding of HRPII to endothelial GAGs induces a pro-inflammatory response, such as disruption of the endothelial barrier by dissociation of cell–cell junctions. Polyphosphates derived from *P. falciparium* and activated platelets bound to HRPII with high affinity and dramatically increased the pro-inflammatory response. The effect of HRPII was competitively counteracted by adding increasing concentrations of AT in vitro [42].

AT is not only active against bacteria, fungi, and parasitic infections with *P. falciparium*, but it also seems to possess anti-viral properties, such as inhibiting cell entry and/or the replication of other viruses such as HSV-1 [43], HIV-1 [44], or various influenza virus isolates, especially against influenza A H1N1 [45]. High-dose AT was even more effective than ribavirin, an antiviral agent, in an in vitro experiment against H1N1. This finding, though, could not be translated to an in vivo mouse model with high-dose intra-nasal application [45]. The authors suggested that the route of AT administration via inhalation would be more effective. However, these experiments were not conducted or have not yet been published [45]. During Hepatitis C virus (HCV) infection, AT, when binding to hepatocytes, inhibits HCV replication via down-regulation of certain genes, which are regulating different signal transduction pathways in the host cells, e.g., of BMP2, CEBPB, and JUN [46].

The antiviral properties of AT are not entirely understood. In HSV-1 infection the heparin-bound AT only inhibited the virus at an entry step [43], but a study with HIV-1 showed that AT induces anti-viral signaling in infected cells, which does not occur in unaffected cells [47]. Elmaleh et al. demonstrated that CD4+ T-cells, CD8+ T-cells, and natural killer cells react to AT by cell migration and inhibit TNF-α-induced NF-κB activation in CD4+ T-cells [48].

Several host cell signal transduction pathways, altered by HIV infection, such as a heat-shock protein pathway, NF-κB pathway, and further pathways, are modulated via treatment with AT, thus, resulting in less virus replication [47]. One central mechanism could be that the prostaglandin-endoperoxide synthase 2 (PTGS2) is activated by AT only in virus-infected cells and that PTGS2 over-expression could significantly reduce HIV virus replication, at least in an in vitro model [47]. This anti-viral effect of heparin-bound AT was independent of prior drug exposure, clade, or co-receptor usage [49].

Regarding the anti-viral effect of AT, the specific type of AT used for the experiments was found to be an important factor. In a primate in vivo model, native AT was administered at supra-physiologic doses, but anti-viral activity, measured by changes in plasma viral RNA levels, could not be found [49]. In contrast, administration of heparin-bound AT, also in a four-fold higher dose of the physiologic concentration, led to an 80% reduction in the plasma viral load [49]. Unfortunately, in connection with this in vivo animal model, no information about side effects of AT, such as bleeding, is reported [49]. In the same study, an in vitro approach was conducted where AT was pre-incubated with heparin. The experiment compared the effect of pre-incubated AT versus heparin alone on human peripheral blood mononuclear cells (hPBMC) that were infected with a human immunodeficiency (HIV) enveloped pseudovirus. In this approach, heparin alone had a comparable inhibitory effect on viral replication as AT-heparin complexes, while AT alone had no effect [49]. Furthermore, other studies found limited antiviral activity of the native form of AT, even if supra-physiologic doses were administered [44,48,49]. If AT is bound to heparin, the required dose is significantly decreased, not only in HIV but also in other viral infections with HCV or HSV [47,48,49]. This might indicate that heparin also has anti-viral properties since it is known that heparin binds to certain viral molecules, such as HIV-1 envelope protein glycoprotein 120 or Herpes simplex virus and dengue virus envelope proteins [50]. However, no information about side effects of the required high doses of AT needed for the anti-viral efficacy, such as bleeding, is reported [49]. As the combination of high doses of AT and the concomitant use of heparin is required for the anti-viral effect of AT, this might be the limiting factor for clinical application as an anti-viral drug in humans. A quite simple, thus, elegant idea is to administer AT in a form that limits interaction with the coagulation system: packaging of AT in immunoliposomes allowed improved targeting of lymphoid organs and, indeed, showed the highest anti-viral results [49].

In addition to direct bacterial killing efficacy, LPS neutralizing, and virus replication inhibition ability of AT, this serpin also has many anti-inflammatory characteristics. Legitimately, high-dose AT administration was, therefore, proposed to attenuate infectious disease syndromes and also proved to be effective in a lethal primate endotoxemia model. In this model, baboons received a lethal dose of *E. coli*. One hour before the bacterial challenge, one group received high-dose AT and the other group received saline solution only. Unfortunately, in this study the bacterial load was not quantified and compared between the treatment groups and so the bactericidal effect of AT could not be directly evaluated. At any rate, all animals in the control group died of multiple organ failure and disseminated intravascular coagulation (DIC), while the animals who received high-dose AT developed less DIC and inflammatory response [37]. Although the decrease in mortality effected by high-dose AT could not be translated to humans, an improvement in organ function and inflammation was detected when administered adjusted to body weight [51]. Therefore, keeping AT on a certain level might contribute positively to the treatment of sepsis by attenuating the pro-inflammatory response due to the anti-inflammatory properties of AT.

## 3. AT and Inflammation

The anti-inflammatory properties of AT can be observed when AT is administered to septic patients; treatment with AT during sepsis led to a decrease in cytokines, such as IL-6, and a decline in CRP concentrations followed by a subsequent drop in body temperature [52]. When heparinized whole blood is treated with AT, the intracellular expression of IL-6 and IL-8 is significantly downregulated in monocytes [53]. While this study did find a slight, not significant, decrease in intracellular TNF-α expression, the TNF-α secretion induced by LPS in human monocytes was significantly reduced by AT in vitro. The authors also found a synergistic effect of AT and Beraprost, a derivative of prostaglandin I_2_, on the suppression of LPS-induced cytokine production by monocytes [54]. When AT is incubated with neutrophils and monocytes, the expression of pro-inflammatory integrins for migration and adhesion is downregulated on activation [55]. The anti-inflammatory activity of AT is conveyed by the D-helix domain of AT, which is also the heparin-binding site [56]. Downstream effects of the anti-inflammatory activity of AT include an inverse correlation between AT and H3 histone levels, and higher levels of AT were associated with reduced immune-mediated tissue damage [57]. H3 histones are cytotoxic and released, for example, by NETosis [58], which can be induced by activated platelets [59]. However, a direct effect of thrombin or antithrombin on neutrophil activation or NETosis, to our knowledge, was never investigated.

Since the initial observation of coagulation-independent anti-inflammatory effects of AT more than 30 years ago [60], two mechanisms explaining the observed anti-inflammatory effects have been confirmed in experiments:*Binding of AT to heparan-sulfate proteoglycans transmits anti-inflammatory effects.* Binding to endothelial heparin-like GAGs induced the release of endothelial prostaglandin I_2_ [61], which inhibits platelet activation and suppresses adhesion and rolling of leukocytes on endothelial cells [62]. AT interacts with the glycosaminoglycan syndecan-4 via its heparin-binding domain [63] and interacts with receptors of the integrin family [64] on the cell surface [65].*AT directly binds to specific cellular receptors that inhibit a pro-inflammatory response.* AT was shown to interact with LRP-1, CD13, and CD300f, which reduces the pro-inflammatory response by suppressing the expression of IL-6, TNF-α, and tissue factor, e.g., by inhibiting formation of the nuclear factor кB (NF-кB) complex [29,30,38,66].

AT levels decrease in inflammatory conditions. The prime reason for the decline in AT plasma concentrations is consumption, which can either be proteolytic cleavage of AT or the less probable spontaneous conformational transformation of an active to a latent form. The molecular characteristics allow AT to enter an energetically more stable conformation upon binding to the target proteases [67,68]. This conversion to a latent form renders the molecule inert. Although latent AT loses its anti-coagulant and anti-inflammatory functions, it was found to act pro-apoptotic and antiangiogenic [69,70]. A likely explanation for this function is that latent AT interferes with the interaction between GAGs and proteins of the extracellular matrix and hampers signal transduction [64]. In an in vitro experiment, the viability of a human kidney cell line (HEK-EBNA cells) was improved by transfection of wildtype and mutant AT but not by supplementation of AT to culture medium, suggesting an intracellular role of AT [71]. A graphical summary of AT and isoform functions is depicted in Figure 1.

## 4. Role of Antithrombin in Tissue Damage

These effects were mostly researched on endothelial and leukocyte cells. As proposed earlier [72], heparan-sulfate proteoglycans are expressed by virtually all cells. Thus, effects of AT on extravascular tissues are theoretically possible. Complete knock-out of AT leads to embryonic lethality with massive subcutaneous hemorrhage and extensive fibrin deposition in the myocardium and the liver [73]. Deletion of one allele of the *Serpinc1* gene, which encodes for antithrombin, led to a higher degree of tissue damage in a liver injury mouse model than did knock-out of anti-apoptotic genes. The effect could have been partially reversed by AT administration [71]. The assumption of extravascular effects is supported by observations in studies that report the effect of AT on ischemia/reperfusion injuries, which is beneficial. An ischemia/reperfusion mouse model testing a mutant form of AT that lacks the thrombin recognition site showed cardioprotective effects with reduced infarct size and plasma levels of troponin I [74]. Another study showed increased hepatic levels of prostaglandin I_2_ after ischemic reperfusion injury in rats [75]. Binding of AT to vascular GAGs preserves the glycocalyx from degradation after reperfusion injury and so maintains the endothelial barrier function, resulting in reduced edema formation [76]. The mechanism by which AT protects the glycocalyx is not fully elucidated. However, there is the theory that binding of AT to heparan-sulfate proteoglycans with its D-helix acts as a shield against proteolytic breakdown [77]. In a rat model challenged with intravenous endotoxin administration, concomitant treatment with a newly developed recombinant AT preparation reduced circulating levels of syndecan-1 and hyaluronan, both components of the glycocalyx. Intravital microscopy revealed a thicker glycocalyx in treated animals, which also had better microvascular perfusion and lower systemic lactate levels [78]. Administration of nebulized AT after chemically induced lung injury in rats showed a reduction of edema, downregulation of tissue factor expression, and consequently, lower fibrin deposition in the pulmonary tissue, an effect that was not observed in the group receiving concomitant heparin [79]. Still, more data are needed to assess the effect of AT on extravascular cells and tissues.

## 5. Antithrombin as a Key Target for Sepsis Treatment

The anticoagulatory and anti-inflammatory properties of AT would seem to make it a promising drug for the treatment of sepsis. There is a non-linear association between AT activity and mortality: the lower the AT concentration, the higher the risk of adverse patient outcome [80]. Notably, AT synthesis is decreased due to liver impairment in septic conditions [81]; in pediatric patients, AT could be a useful prognostic marker for predicting organ dysfunction and mortality in sepsis [13]. Recent trials state that treatment with AT had a positive effect on inflammatory processes, among which are reduced NETosis [82]. Treatment with AT was associated with improved recovery from disseminated intravascular coagulopathy (DIC) [82,83,84], even when concomitant anticoagulants were given [83,85,86].

In a retrospective trial conducted in 926 patients with sepsis-induced DIC, Iba et al. calculated a cutoff value for death, namely, for AT activity at baseline at about 41% and for an AT activity level after AT administration at about 71% and, thus, showed that activity-monitored AT supplementation has a beneficial impact on the survival rate [87]. The most well-known clinical study investigating the effects of AT in septic patients is the KyberSept trial [88]. Although the study included a large number of patients (2314 individuals), the trial failed to prove a significant difference in the survival rate in the AT group versus the placebo group but demonstrated significantly more bleeding events in the AT group. The study undermined the critical role of heparin treatment: subgroup analysis revealed that patients who did not receive concomitant heparin had a survival benefit and lower rates of bleeding events, while survival of patients who suffered major bleeding did not differ between the groups [88]. A subgroup analysis revealed a reduction in mortality in patients with DIC not receiving concomitant heparin [89]. A subsequent meta-analysis found no difference in the mortality of septic patients with and without DIC but, again, detected an increased bleeding incidence following the administration of AT [90]. A notable limitation of the meta-analysis is the fact that KyberSept contributed the large majority of meta-analyzed patients, thus, biasing the results towards the outcome of the KyberSept trial, which did not primarily stratify included patients according to the presence of DIC [91].

The non-systemic administration of AT might provide a reasonable form of application. Nebulized AT was associated with limited bacterial growth and reduced pulmonary coagulopathy in rats challenged intratracheally with *S. pneumonia* [41]. Another potential therapeutic use of anticoagulants was discussed for patients suffering from smoke-inhalation-associated acute lung injury, a serious injury seen in fire victims. Although only data from animal studies are available, application of nebulized AT resulted in promising improvements in clinical endpoints, such as days on mechanical ventilation or PaO2/FiO2 after weaning, without altering systemic coagulation parameters [92].

Nevertheless, a potential beneficial effect of AT supplementation, particularly regarding the evaluation of further clinical outcomes of sepsis in different patient groups, is discussed. Subgroup analyses and retrospective trials provided evaluations of effects beyond the hard outcome of “mortality”. This included, i.e., a reduced time to recovery from DIC [93] by improvement of platelet counts [84]. AT levels inversely correlate with the risk of sepsis-induced AKI [14]. Another trial, investigating the efficacy of high-dose AT supplementation in septic patients with DIC, showed improved survival for the patient group that received high-dose supplementation of AT. Unfortunately, the study lacked a control group receiving placebo [94]. Kim et al. specifically investigated adult septic shock patients suffering from DIC and AT levels of <70% and were able to show a reduced 28-day mortality when AT was supplemented [95]. AT supplementation in sepsis-induced DIC is common clinical practice in Japan and recommended specifically in patients whose AT levels dropped below 70% [96]. Recently, rAT-gamma, which lacks a core fucose complex, thus, rendering it less affine to procoagulant enzymes, was approved by Japanese authorities for therapeutic use in patients with congenital AT deficiency and in sepsis-induced DIC [97]. In summary, the current data suggest a potentially beneficial effect in patients with sepsis-induced DIC and no effect, or even a harmful effect, in unselected critically ill patients. In order to break the scientific stalemate, AT supplementation should be tested as targeted therapy in study cohorts, where recent data suggest that a specific patient population could benefit from the therapeutic supplementation [98].

## 6. Immunologic Implications of Therapeutic Antithrombin Supplementation

Despite the potential positive effects, AT supplementation is controversially discussed. A major concern is bleeding risk after administration, especially with the concomitant administration of heparin, a frequent preventive therapeutic measure in septic patients; the heparin-binding site of AT and the binding site of AT for its anti-inflammatory interactions are identical. A potentially competitive reduction of the anti-inflammatory capacity of AT was shown in a hamster window model with LPS-induced sepsis, in which microvascular leukocyte–endothelial (LE) cell interactions and alterations in functional capillary density (FCD) were neutralized to levels measured in untreated hamsters as soon as AT was given together with UFH or LMWH [65,99]. This abrogating effect was also detected in a subgroup analysis in the KyperSept trial, where only patients not receiving concomitant heparin showed a survival benefit [88]. To prevent the inhibition of beneficial immune-modulatory and antimicrobial effects of AT by heparin, alternative anticoagulatory substances, such as argatroban, can be used. Argatroban is a well-known treatment alternative for patients suffering from heparin-induced thrombocytopenia (HIT) and exhibits its anticoagulatory effects via the direct inhibition of thrombin [100]. It is a small molecule and binds reversibly to the catalytic site of thrombin. Argatroban exerts its anticoagulatory effects on soluble and clot-bound thrombin and has an almost linear dose–response relationship. The short half-life of about 40 min makes it a well-controllable substance [101,102]. In addition to its direct anticoagulatory effect, thrombolytic properties have been proven in patients suffering from thrombotic ischemic events, such as myocardial infarction [103]. Direct thrombin inhibition could have further beneficial effects since a main characteristic of DIC is upregulated thrombin formation [102]. While the data on patients suffering from DIC using argatroban are insufficient, small case series suggest a beneficial effect of argatroban on recovery from DIC [104,105]. Animal models support these promising effects of direct thrombin inhibitors in the prevention of DIC [106]. Critically ill patients who suffer from heparin resistance might also benefit from anticoagulation with argatroban. Target activated partial thromboplastin times (aPTT) can be achieved faster and more easily with argatroban than with heparin (Bachler et al.) [107]. Critically ill patients who suffer from sepsis and DIC are a very vulnerable patient group and target-specific treatments are critically needed. Low levels of AT are common in sepsis or other conditions with a severely inflammatory phenotype, such as COVID-19, where thrombin generation markers are high and standard treatment with LMWH is ineffective [108]. Based on these observations, it is suggestive that medications, such as direct thrombin inhibitors, should be preferred over treatment regimens that are known to interfere with immunologic pathways, e.g., heparin. However, clinical studies proving this speculation are lacking.

## 7. Conclusions

Although the available data are currently still insufficient to facilitate solid clinical decision-making, the potential of AT in sepsis-induced DIC should be further investigated. Treatment options for sepsis and, especially, DIC are sparse and have not improved significantly over recent years; the research of AT as an anti-inflammatory, immune-modulating and antimicrobial drug still holds potential. Discovery of the distinguished roles of the α and β isoforms of AT might provide the advance needed to exploit the beneficial effects of AT without increasing the risk for major bleedings or might help identify patients who would benefit most from AT supplementation. Theoretically effective treatment options of the anti-inflammatory mechanisms conveyed by AT remain to be evaluated in experiments. Also, a comparison of sepsis severity groups, with or without DIC, in the scale of the KyberSept trial would be of great interest. Until then, AT supplementation in septic patients with confirmed DIC and reduced AT levels might exert beneficial effects on inflammatory processes and survival, but it remains an off-label use.

## Figures and Tables

**Figure 1 ijms-22-04283-f001:**
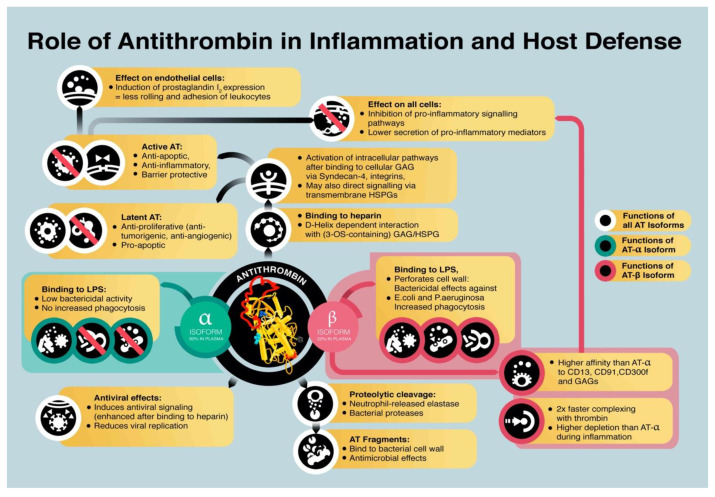
Functions and effects of antithrombin (AT) and AT isoforms in host defense and inflammation. LPS = lipopolysaccharide; HSPG = heparan sulfate proteoglycans; GAG = glycosaminoglycans.

## Data Availability

Not applicable.

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
