# Peer review of "Antithrombin and Its Role in Host Defense and Inflammation"

_ijms, 2021, doi:10.3390/ijms22084283_

Round 1
Reviewer 1 Report
This is a review that focuses on the roles of antithrombin in host defense and inflammation, which is very similar to last years’ publication: “Antithrombin: An anticoagulant, anti‐inflammatory and antibacterial serpin” 28 February 2020 jth.
For readers, the authors shall give much more details on the structure and function of antithrombin molecule as well as the roles in coagulation cascade and signaling pathways during host defence and inflammation.
The diagram is a bit too complicate. It would be easy for reader if the authors could change the presentation. The reference used are more old ones than recent ones. It would be better to include more recent reference including Antithrombin in COVID-19.
Author Response
Dear Editors, dear referee,
we thank you for considering our review for publication, and especially the reviewers for the critical appraisal of our manuscript – the time it takes and for the suggestions - and hope we could improve it accordingly. The manuscript was also proof-read by a native speaker to eliminate language deficits. All changes made (except for spelling or grammatical corrections) are highlighted in yellow.
This is a review that focuses on the roles of antithrombin in host defense and inflammation, which is very similar to last years’ publication: “Antithrombin: An anticoagulant, anti-inflammatory and antibacterial serpin” 28 February 2020jth. For readers, the authors shall give much more details on the structure and function of antithrombin molecule as well as the roles in coagulation cascade and signaling pathways during host defence and inflammation.
Answer: We agree that the topic of this review is very similar, the referenced review is an excellent work which particularly covers the most important findings regarding the molecular mechanisms of AT in inflammation and host response. By putting an emphasis on AT in clinical applications, we tried to add other aspects and shift the focus within this topic. We added more details regarding the role and function of AT in the coagulation cascade, however we think that the review of Rezaie et al. (and another review from the same author: “Anticoagulant and signaling functions of antithrombin” 11 Aug 2020 jth) very thoroughly summarizes the molecular pathways during host response and inflammation, so we decided it would be more fair to add the review as citation to direct the reader to these excellent articles (references 9 and 25).
The diagram is a bit too complicate. It would be easy for reader if the authors could change the presentation.
Answer: We adopted the illustration: we added coloured elements to visually structure the diagram according to the colour coding of the items. We also reduced the amount of text in the illustration to reduce the perceived load of information.
The reference used are more old ones than recent ones. It would be better to include more recent reference including Antithrombin in COVID-19.
Answer: We very much appreciate the reviewers suggestion to include data about COVID-19 and antithrombin in our review and inserted a paragraph describing recent findings regarding AT in COVID-19 patients.

Reviewer 2 Report
Review
Manuscript ID: ijms-1110823
Type of manuscript: Review
Title: Antithrombin and its role in host defense and inflammation
Authors: Christine Schlömmer, Anna Brandtner *, Mirjam Bachler
Overall, the authors provide a sound and well prepared overview of antithrombin in anti-infalmmatory and cytoprotective processes.
My major remark would be that given the more basic scientific character of the topic and the composed references, the conclusion focuses too much on the direct clinical appliance of AT.
I feel that some minor changes and clarifications might also benefit the message of this review article.
- 1 ll. 38-41: I can not see how the fact that AT levels and their correlation to unfavourable outcomes in septic patients is due to the primary role of AT as an anticoagulant. Furthermore, the TAT complexes are not addressed further on.
- 2 ll. 65-66: This statement is too speculative and does not seem to be fully supported by the undelying references.
- 3 l. 128: Too speculative
- 3 ll. 136-142: The authors should discuss the possible role of heparin alone in this in vivo model.
- 4 l. 156: Erase [18]
- 4 l. 182: „include an inverse correlation between AT and H3 histone levels“: Please provide further details how this contributes tot he anti-inflammatory properties of AT.
- 5: Figure 1 should be captioned „Role of Antithrombin in inflammation and host defense“ within the graph.
- 6 ll. 245-246: The remark on high bilirubin levels and laboratory testing does not add anything tot he main topic of this passage.
- 6 ll. 247-251: This sentence needs editing.
- 7 l.304: „with heparin probably diminishing the beneficial effects of AT“ Can the authors provide primary sources for this assertion.
- 7 ll. 327-330: The claim that citically ill patients „are a very fragile biological system“ seems a little inapt. The claim that direct anticoagulants should be preferred over heparin in these patients needs further explanation.
Author Response
Dear Editors, dear referee,
we thank you for considering our review for publication, and especially the reviewers for the critical appraisal of our manuscript – the time it takes and for the suggestions - and hope we could improve it accordingly. The manuscript was also proof-read by a native speaker to eliminate language deficits. All changes made (except for spelling or grammatical corrections) are highlighted in yellow.
Overall, the authors provide a sound and well prepared overview of antithrombin in anti-infalmmatory and cytoprotective processes.
My major remark would be that given the more basic scientific character of the topic and the composed references, the conclusion focuses too much on the direct clinical appliance of AT.
I feel that some minor changes and clarifications might also benefit the message of this review article.
- 1 ll. 38-41: I can not see how the fact that AT levels and their correlation to unfavourable outcomes in septic patients is due to the primary role of AT as an anticoagulant. Furthermore, the TAT complexes are not addressed further on.
Answer: We removed the statement with TAT complexes as there is no connection in the following, as the reviewer correctly pointed out.
- 2 ll. 65-66: This statement is too speculative and does not seem to be fully supported by the undelying references.
Answer: We thank the reviewer very much for this attentive comment: indeed we missed to introduce the correct reference: this was updated and further publications were included. We also corrected and modulated the statement to underline the hypothetical character of this theory.
- 3 l. 128: Too speculative
Answer: We removed the corresponding statement.
- 3 ll. 136-142: The authors should discuss the possible role of heparin alone in this in vivo model.
Answer: The reviewer correctly pointed out an opportunity for addition of interesting information. We added a paragraph which discusses the effects of heparin alone in this model.
- 4 l. 156: Erase [18]
Answer: We erased reference [18] at this position.
- 4 l. 182: „include an inverse correlation between AT and H3 histone levels“: Please provide further details how this contributes to the anti-inflammatory properties of AT.
Answer: We very much thank the reviewer for this careful note: we indeed did not find any further work which analysed the effect of antithrombin or thrombin on the activation of neutrophils or NETosis. We added a corresponding statement to the section.
- 5: Figure 1 should be captioned „Role of Antithrombin in inflammation and host defense“ within the graph.
Answer: We adapted the caption of the illustration accordingly.
- 6 ll. 245-246: The remark on high bilirubin levels and laboratory testing does not add anything to the main topic of this passage.
Answer: We agree and deleted the passage.
- 6 ll. 247-251: This sentence needs editing.
Answer: The reviewer correctly points out that the sentence, which spans from line 247-251 is confusing. We modulated the statement and increase the readability.
- 7 l.304: „with heparin probably diminishing the beneficial effects of AT“ Can the authors provide primary sources for this assertion.
Answer: The reviewer made a valuable suggestion to improve this statement. We changed the wording and connected it more directly to the primary source for this assertion.
- 7 ll. 327-330: The claim that critically ill patients „are a very fragile biological system“ seems a little inapt. The claim that direct anticoagulants should be preferred over heparin in these patients needs further explanation.
Answer: We replaced this line with a more appropriate wording and also reworded the preference of direct thrombin inhibitors over heparin.

Round 2
Reviewer 1 Report
Getting better now